# Complete Genome Sequence of Two Deep-Sea *Streptomyces* Isolates from Madeira Archipelago and Evaluation of Their Biosynthetic Potential

**DOI:** 10.3390/md19110621

**Published:** 2021-11-01

**Authors:** Pedro Albuquerque, Inês Ribeiro, Sofia Correia, Ana Paula Mucha, Paula Tamagnini, Andreia Braga-Henriques, Maria de Fátima Carvalho, Marta V. Mendes

**Affiliations:** 1i3S—Instituto de Investigação e Inovação em Saúde, Universidade do Porto, Rua Alfredo Allen 208, 4200-135 Porto, Portugal; pedro.albuquerque@i3s.up.pt (P.A.); pmtamagn@ibmc.up.pt (P.T.); 2IBMC—Instituto de Biologia Molecular e Celular, Universidade do Porto, Rua Alfredo Allen 208, 4200-135 Porto, Portugal; 3CIIMAR—Interdisciplinary Centre of Marine and Environmental Research, University of Porto, Terminal de Cruzeiros do Porto de Leixões, Avenida General Norton de Matos s/n, 4450-208 Matosinhos, Portugal; iribeiro@ciimar.up.pt (I.R.); sofia_fariacorreia@hotmail.com (S.C.); amucha@ciimar.up.pt (A.P.M.); mcarvalho@ciimar.up.pt (M.d.F.C.); 4ICBAS—Instituto de Ciências Biomédicas Abel Salazar, Universidade do Porto, Rua de Jorge Viterbo Ferreira 228, 4050-313 Porto, Portugal; 5Departamento de Biologia, Faculdade de Ciências, Universidade do Porto, Rua do Campo Alegre, Edifício FC4, 4169-007 Porto, Portugal; 6OOM—Oceanic Observatory of Madeira & MARE—Marine and Environmental Sciences Centre, ARDITI—Agência Regional para o Desenvolvimento da Investigação Tecnologia e Inovação, Caminho da Penteada, 9020-105 Funchal, Portugal; braga.henriques02@gmail.com; 7Regional Directorate for Fisheries, Regional Secretariat for the Sea and Fisheries, Government of the Azores, Rua Cônsul Dabney—Colónia Alemã, 9900-014 Horta, Portugal

**Keywords:** *Streptomyces*, deep-sea actinobacteria, de novo assembly, genome mining, natural products

## Abstract

The deep-sea constitutes a true unexplored frontier and a potential source of innovative drug scaffolds. Here, we present the genome sequence of two novel marine actinobacterial strains, MA3_2.13 and S07_1.15, isolated from deep-sea samples (sediments and sponge) and collected at Madeira archipelago (NE Atlantic Ocean; Portugal). The de novo assembly of both genomes was achieved using a hybrid strategy that combines short-reads (Illumina) and long-reads (PacBio) sequencing data. Phylogenetic analyses showed that strain MA3_2.13 is a new species of the *Streptomyces* genus, whereas strain S07_1.15 is closely related to the type strain of *Streptomyces xinghaiensis*. In silico analysis revealed that the total length of predicted biosynthetic gene clusters (BGCs) accounted for a high percentage of the MA3_2.13 genome, with several potential new metabolites identified. Strain S07_1.15 had, with a few exceptions, a predicted metabolic profile similar to *S. xinghaiensis*. In this work, we implemented a straightforward approach for generating high-quality genomes of new bacterial isolates and analyse in silico their potential to produce novel NPs. The inclusion of these in silico dereplication steps allows to minimize the rediscovery rates of traditional natural products screening methodologies and expedite the drug discovery process.

## 1. Introduction

Historically, natural products (NPs) have been a valuable source of chemical scaffolds for the drug discovery pipeline. Traditional screening methodologies for microbial-derived NPs are becoming obsolete as they rely on the ability of the microorganism to produce the metabolite in laboratory conditions. In addition, the outcome frequently leads to the rediscovery of known compounds, highlighting the importance of the implementation of dereplication strategies in NP screening workflows [1]. 

The technical advances brought by the post-genomic era led to an accumulation of fully sequenced bacterial genomes in the databases [2,3]. With the scrutiny of these genomes, it became clear that bacteria harbour in their genome an untapped potential for the production of novel NPs. This is particularly true for bacteria of the *Streptomyces* genus. Genome-wide studies show that *Streptomyces* genomes can harbour on average more than 20 biosynthetic gene clusters (BGCs) for the production of NPs but only a small fraction of these is produced under standard laboratory conditions [4,5,6]. Activation of those BGCs that have reduced expression or are not expressed at all has emerged as a key strategy for the identification and production of novel bioactive compounds [7]. A key step prior to BGC activation is mining bacterial genomes for genes that are likely to govern the biosynthesis of NP scaffold structures. As bioinformatics tools evolve, genome mining is becoming an increasingly effective strategy for in silico dereplication of microbial metabolites and expediting the BGC activation workflow.

Genome mining and accurate in silico identification of BGCs requires high-quality genomes sequences [8]. Short-read sequencing technologies such as Illumina, are widespread, low-cost, present high coverage, and deliver high fidelity reads [9]. However, the exclusive use of short-reads data for de novo assembly of complex bacterial genomes can lead to incomplete assemblies due to the presence of repetitive elements or genome duplications [10]. Long-reads technologies, such as PacBio, have improved the accuracy of de novo assembly by providing information regarding the genomic structure. Although long-reads are characterized by a greater sequencing error rate compared to the Illumina sequencing [9,11,12], their higher read length increases the contiguity of the assembly and prevents errors due to the presence of duplicated and/or repetitive regions [13]. The use of hybrid strategies for de novo assembly of complex bacterial genomes combines the accuracy of short reads with information of the genomic structure provided by the long reads [14]. These hybrid strategies have proven to be particularly efficient for the assembly of high GC genomes with a high incidence of repetitive sequences such as those of streptomycetes [15]. 

Genome mining of actinomycetes from marine environments, including deep-sea, has emerged as a key approach for the identification of new compounds [16,17]. Due to the extreme environmental conditions deep-sea derived actinomycetes, in particular, *Streptomyces*, display unique metabolic features leading to the production of NPs with distinctive chemical structures and bioactivities [18,19]. Here we report the de novo high-quality sequence of two *Streptomyces* genomes, including one novel *Streptomyces* species, isolated from deep-sea samples. The potential of the two isolates to produce novel NPs was evaluated through an in-depth bioinformatics analysis of each genome.

## 2. Results

### 2.1. Isolation, Phenotypic Characterization and Sequencing

Isolates MA3_2.13 and S07_1.15 were isolated from samples collected during two oceanographic expeditions in the Madeira archipelago (NE Atlantic Ocean; Portugal). Isolate MA3_2.13 grew on M1 medium after 2 months of incubation at 28 °C and the colonies presented a brownish vegetative mycelium and a white aerial mycelium. Isolate S07_1.15 was retrieved from M4 medium after an incubation period of 3 months and its colonies presented a whitish vegetative mycelium and a white/grey aerial mycelium.

A BLAST (blastn) analysis of the 16S rRNA partial sequences obtained by PCR, showed a sequence similarity of 99.07% of isolate MA3_2.13 with *Streptomyces* sp. NPS-554 [20] and isolate S07_1.15 presented 100% identity with *Streptomyces xinghaiensis* S187 [21]. Both of these strains were reported to be isolated from marine sediments. To further characterize isolates MA3_2.13 and S07_1.15 we fully sequenced and analysed in silico their genomes by implementing multiple phylogenetic analyses on the basis of the 16S rRNA sequences, single-copy core genes and whole-genome sequences (WGS). 

The genomic DNA of both isolates was used for the generation of Illumina and PacBio sequencing libraries. After quality control and filtering of raw reads, Illumina sequencing generated a total of 7,097,472 (139× coverage) and 7,868,594 (163×) high-quality paired-end reads for isolate MA3_2.13 and S07_1.15, respectively. Of these, 92.06% and 91.94% of the reads of isolate MA3_2.13 and S07_1.15, respectively, presented an average Phred score of Q30 or higher. Sequencing of PacBio libraries generated 76,363 high-quality subreads (N50-10429 nt) for isolate MA3_2.13 (85-fold coverage) and 61,119 high-quality subreads (N50-10555 nt) for isolate S07_1.15 (69-fold coverage).

### 2.2. Genome Assembly and Annotation

The de novo genome assembly of the two isolates was generated by combining PacBio and Illumina sequencing data using the Unicycler workflow [9], followed by manual curation via mapping the Illumina reads in the originated contigs. The genomic features of the two isolates are summarized in Table 1.

The genome assembly of isolate MA3_2.13 generated a unique contig of 7.7 Mbp with an average G+C content of 72.1% (Figure 1A). RAST [22] annotation identified 6412 CDS, 5 ribosomal RNA operons and 55 tRNAs. Analysis with BUSCO (v. 5.0.0) [24] (actinobacteria_class_odb10 ortholog set), revealed the presence of 290 out of 292 (99.3%) actinobacterial core genes of which 287 were found in a single copy and 3 were duplicated. CRISPRCasFinder [25] analysis showed that the genome of MA3_2.13 harbors 3 type I *cas* operons and 7 CRISPR arrays.

The assembly graph of isolate S07_1.15 reveals two linear contigs of 7.1 Mbp (average G+C content 73.2%) (Figure 1B) and 160,397 bp (average G+C content 69.6%). Analysis of the Illumina reads mapping revealed an increase in the average coverage of the 160 kb fragment compared to the larger contig (226-fold vs. 158-fold) which suggest either the presence of an extrachromosomal replicon or chromosomal repeated regions that could not be assembled into the 7.1 Mbp contig. A total of 6671 CDS, 6 ribosomal RNA operons, and 63 tRNAs were identified by RAST annotation, and 290 out of 292 (99.3%) actinobacterial core genes were found of which 289 in a single copy and 1 duplicated. Two type I *cas* loci and 4 CRISPR arrays were identified in the genome of isolate and S07_1.15. 

Comparison of the functional annotation of predicted genes (Appendix A) revealed different Clusters of Orthologous Groups (COGs) abundance patterns between MA3_2.13 and S07_1.15. Particularly, strain MA3_2.13 harbours fewer genes related to signal transduction mechanisms (T), when compared to both S07_1.15 and well-studied *Streptomyces* (*S. coelicolor, S. avermitilis* and *S. griseus*). Inversely, strain S07_1.15 has less proportion of genes related to transcription (K), when compared to other *Streptomyces*. Concerning metabolism-related categories, strain MA3_2.13 generally exhibits a higher proportion of genes assigned to the eight categories (42% in MA3_2.13 versus 35.6% in S07_1.15), with particular emphasis for genes related to carbohydrate transport and metabolism (G), inorganic ion transport and metabolism (P), and secondary metabolites biosynthesis, transport, and catabolism (Q). Furthermore, when compared to well-studied *Streptomyces,* the proportion of genes in categories G, P and Q are either similar or higher in strain MA3_2.13.

Both genome assemblies showed several regions that were predicted as putative genomic islands. In strain MA3_2.13, 17 regions were annotated as putative islands and in strain S07_1.15 a total of 29 regions. In both assemblies, the regions were distributed across the entire length of the chromosome. Although a few predicted islands corresponded to annotated CRISPR arrays, in many instances (12 out of 17 in MA3_17 and 11 out of 29 in S07_1.15) these regions contained annotated genes linked to genomic instability namely transposases, mobile elements and integrases (Appendix A). A complementary search for prophage sequences in both genomes revealed the presence of one region in strain MA3_2.13 spanning 18.9 Kb that is very rich in prophage sequences and includes the flanking attachment site junctions *attL* and *attR*. This region was considered as a genomic island in the above-mentioned analysis. In the case of strain S07_1.15, a total of 8 regions contained several prophage genes, with region sizes ranging from 6.2Kb to 11.1 Kb (Appendix A).

### 2.3. Phylogenetic Analysis of the Deep-Sea Isolated Strains

The 16S rRNA phylogenetic analysis (Appendix A) showed that isolate MA3_2.13 clustered together with two other *Streptomyces* strains isolated from marine environments: *Streptomyces* sp. NPS-554 [20] and *Streptomyces* sp. CNQ-233 SD01 [26]. However, the branch of the tree supporting this cluster had low bootstrap support. Isolate S07_1.15 strongly clustered with *Streptomyces xinghaiensis*, among other *Streptomyces* sp. isolated from samples collected in the Yellow Sea (e.g., *Streptomyces* sp. FXJ7.369, *Streptomyces* sp. A165 and *Streptomyces* sp. FXJ7.368). 

For better phylogenetic resolution, we performed a multi-locus sequence analysis (MLSA), using the concatenated sequences of five housekeeping genes (*atpD*, *gyrB*, *recA*, *rpoB* and *trpB*) (Appendix A). This analysis showed that isolate MA3_2.13 did not cluster closely with any other included strain, which potentiates its status as a novel *Streptomyces* species. On the other hand, isolate S07_1.15, even though closely related to *S. xinghaiensis*, had its strongest clustering with *Streptomyces* sp. WAC 00631, a soil isolate from Canada. 

Taking advantage of the obtained full genome sequences of the two isolates, we additionally carried out whole-genome phylogenetic analyses. We started by using the *NCBI Prok* query of the Microbial Genomes Atlas (MiGA) (v. 0.7.15.2) webserver [27]. The closest taxonomic relatives reported for isolate MA3_2.13 were *Streptomyces* sp. SCSIO 3032 (GenBank Assembly accession GCA_002128305 [28]; *p*-value: 0.91) with 67.47% average amino acid identity (AAI) and *Streptomyces harbinensis* NA02264 (GenBank assembly accession GCA_013364095; *p*-value: 0.926) with 66.31% AAI. The results further suggest that isolate MA3_2.13 belongs to the *Streptomyces* genus (*p*-value: 0.34), although likely to a species not represented in the NCBI database (*p*-value: 0.0021). For isolate S07_1.15, the closest reported relative was *Streptomyces xinghaiensis* S187 (GenBank assembly accession GCA_000220705 [29]; *p*-value: 0.049) with an ANI value of 96.66% (AAI of 95.7%). The results from the MiGA server matched with the KmerFinder best scores obtained for both isolates: *Streptomyces* sp. SCSIO 03032 (score = 4924), isolated from deep-sea sediment from the Indian Ocean [28] for isolate MA3_2.13 and *Streptomyces xinghaiensis* S187 (score = 63,434) for isolate S07_1.15. 

We generated a phylogenetic tree with all available (as of March 2021) NCBI RefSeq complete *Streptomyces* sp. genomes (280 genomes) and the genomes of our two isolates. The tree was constructed using the single-copy gene HMM profile specific to the Actinobacteria phylum (138 genes) (Figure 2) [30]. The obtained WGS tree showed that the most closely related strains to isolates MA3_2.13 and S07_1.15 were *Streptomyces* sp. SCSIO 3032 (GCF002128305) and *S. xinghaiensis* S187 (GCF000220705), respectively. Moreover, both isolates clustered with additional strains isolated from marine-derived samples. The whole-genome average nucleotide identity (ANI) between the closest strains (selected based on the pairwise distance matrix of the WGS-based tree) was determined (Appendix A). For isolate MA3_2.13, the top ANI value was 77.90% with *Streptomyces* sp. SCSIO 3032, which is below the 95–96% threshold recommended for prokaryotic species delineation [31,32]. The ANI value for isolate S07_1.15 with *S. xinghaiensis* S187 was 95.83%.

### 2.4. Marine Adaptation Genes

In general, genes coding for ABC transporters, potassium and sodium transporters, genes related to transcriptional regulation and to electron transport, were reported as potential adaptations to the marine environment and as such considered as Marine Adaptation Genes (MAGs) for *Salinospora* and *Streptomyces* strains [33,34,35]. The search of putative MAGs in our two assemblies showed that from a total of 107 analysed genes (retrieved from the MAG lists of three previous works [33,34,35]) the genome of MA3_2.13 contained 38 of these genes while strain S07_1.15 contained 35. Among these, several ABC transporters, ion transporters (namely Na^+^ and K^+^) and transcriptional regulators are included (Appendix A). In the case of the *nuo* operon (respiratory complex I, NADH:ubiquinone oxidoreductase), previous works consistently detected an extra copy in marine strains, when compared to terrestrial strains, speculating that the encoded proton pump helps maintain a proton gradient in seawater [35]. Terrestrial or marine strains displayed both a complete *nuoABCDEFGHIJKLMN* operon and a partial *nuoABCHIJKLMN* operon. In addition, marine genomes also displayed a partial *nuoAHJKLMN* operon which was considered as MAGs [35]. Interestingly, this additional *nuoAHJKLMN* operon was only found in strain S07_1.15, with strain MA3_213 only containing the *nuoABCDEFGHIJKLMN* and *nuoABCHIJKLMN* operons.

### 2.5. Secondary Metabolism in Silico profiling

To assess the secondary metabolite biosynthetic potential of both isolates we analysed the genomes with antiSMASH [23]. A total of 32 and 24 BGCs, classified according to antiSMASH cluster types, were identified for isolates MA3_2.13 (Appendix A) and S07_1.15 (Appendix A), respectively. The total length of the BGCs accounted for 23.1% and 8.8% of the genome of MA3_2.13 and S07_1.15, respectively, which in the case of strain MA3_2.13 is considered to be a high proportion of the genome devoted to secondary metabolites [36]. Only ca 30% of the identified BGCs in both genomes showed gene homologies with known clusters at the MIBiG database [37]. These included common secondary metabolites BGCs found in *Streptomyces* such as ectoine, hopene, desferrioxamine, SapB and geosmin [38]. A comparison between both isolates (Figure 3) showed that 53% (17 out of 32) of the BGCs identified in the genome of MA3_2.13 are devoted to the biosynthesis of polyketide-based metabolites, whereas in S07_1.15 only two polyketide-based encoding clusters were identified, namely a type II PKS and a type III PKS. Interestingly, no type I PKS BGC was identified in S07_1.15. On the other hand, the proportion of RiPPs-encoding BGCs in S07_1.15 is higher when compared to MA3_2.13 (42% vs. 19%).

Regarding isolate MA3_2.13, in silico analysis showed that 52% of the genes within the BGC #8 had a significant BLAST hit with the atratumycin BGC (*atr* cluster; MIBiG accession number BGC0001975) from *Streptomyces atratus* SCSIO ZH16, isolated from deep-sea sediment samples [39]. Atratumycin is a cyclic decadepsipeptide synthesized by 3 NRPS encoding genes, with an N-terminal cinnamoyl acid moiety that displays activity against *Mycobacteria tuberculosis* [39,40]. A careful analysis of the MA3_2.13 BGC #8 showed a similar gene organization when compared to atratumycin BGC namely the presence of three NRPS encoding genes that are interestingly located within a predicted genomic island. Despite the gene synteny of BGC #8 with the atratumycin BGC, the sequence identities between the NRPS proteins vary between 49% and 57% suggesting the biosynthesis of a novel cyclic decadepsipeptide. The three NRPSs from BGC #8 are predicted to assemble a 10 amino acid core backbone with the following sequence: L-Thr, L-Asn, L-orn, D-Ser, L-Phe, L-Pro, D-Val, L-Gly, D-orn, L-Gly. The presence of epimerization (E) domain in modules 4, 7 and 9 is noteworthy and suggests the incorporation of D-amino acids. Downstream from the NRPS encoding genes is located a set of 14 encoding genes that display sequence identity with Atr5 to Atr16 proteins from the atratumycin BGC, suggesting their involvement in the biosynthesis of the cinnamoyl moiety. 

The BGC #14 from MA3_2.13 displayed significant similarity with the triacsins BGC (*tri* cluster) from *Kitasatospora aureofaciens* (MIBiG accession number BGC0001983) [41]. Triacsins are inhibitors of the acyl-CoA synthetase characterized by an 11-carbon unsaturated alkyl chain and an N-terminal N-hydroxytriazene moiety. In silico analysis of BGC #14 revealed the presence of the *tri* homologs for the PKS-related encoding genes, putatively involved in the biosynthesis of the unsaturated alkyl chain [41]. The search for genes involved in the N-N bond formation of the N-hydroxytriazene moiety retrieved homologues for the CreE/Tri21 and CreM/Tri19 proteins although no homolog for CreD/Tri16 was found. These proteins are involved in N-N bond formation in cremeomycin [42] and triacsins, respectively. Instead, two genes were identified that encoded homologues to Spb39 and Spb40 proteins implicated in the biosynthesis of hydrazinoacetic acid, a putative precursor for the hydrazone unit of s56-p1, a dipeptide produced by *Streptomyces* sp. SoC090715LN-17 [43]. This result suggests that the biosynthesis of the N-hydroxytriazene in BGC #14 could involve a novel mechanism of N-N bond formation. Interestingly, CreE and CreD homologs were identified within BGC #9 of MA3_2.13 coding for a type III PKS.

A remarkable feature of the MA3_2.13 genome is that 13 BGCs are predicted to code for type I PKS, either as single PKS clusters or hybrid NRPS-PKS clusters, and only 4 BGCs displayed similarities with known clusters (BGC #24, #29, #31 and #32). In the case of hybrid NRPS-PKS clusters #24, #31 and #32, the gene similarities between clusters were limited to the PK regions. For instance, 79% of genes from BGC #24 showed similarity with an arseno-polyketide from *S. lividans* 1326 (MIBiG accession number BGC0001283) [44]. In addition to the unimodular PKS encoding gene homolog to SLI_1088, homologs for the three genes responsible for the As-C bond were identified in BGC #24. However, no similarities were identified for the NRPS region. This result suggests that cluster BGC #24 either might code for a novel hybrid arseno-metabolite or be split into two neighbouring clusters: a PKS and a NRPS. Likewise, gene organization of clusters BGC #23, #24 and #31 can also raise some doubts regarding their hybrid nature. Nevertheless, in the case of hybrid NRPS-PKS BGCs #1, #6, #19, #27, #30 and #32 they seem to be true hybrid clusters as the two sub-cluster regions are interleaved.

Among the type I PKS encoding clusters, two clusters (BGC #2 and #19) present a monomodular PKS encoding gene that showed identity with iterative type I PKS (iT1PKS). The presence of iT1PKS in *Streptomyces* is more common and widespread than initially predicted and are responsible for the biosynthesis of complex products such as allenic polyketides and citreodiols [45]. Most notably, BGC #19 harbours a hybrid iT1PKS/NRPS encoding gene that displays 68% identity with IkaA of the polycyclic tetramate macrolactam ikarugamycin BGC [46]. Finally, it should be highlighted the presence of BGC #18, a type 1 PKS with 25 modules which constitutes one of the largest PKS assembly lines [47].

Concerning isolate S07_1.15, the phylogenetic analysis showed that this strain is closely related with *S. xinghaiensis* S187 (=NRRL B-24674), isolated from marine sediment and whose biosynthetic potential was previously analysed [21,29]. In silico genome mining showed a very similar BGC profile between S07_1.15 and *S. xinghaiensis* as most of the BGCs showed significant gene similarities with clusters from *S. xinghaiensis* (Appendix A). Nonetheless, homologous clusters to Pks1, Nrps2 and Lan3 BGCs from *S. xinghaiensis* were not identified in the genome of S07_1.15. Inversely, 4 BGC from S07_1.15 (BGC #12, #16, #18 and #22) showed no counterparts in the *S. xinghaiensis* genome. 

A distinctive feature of *S. xinghaiensis* is its ability to produce fluoroacetate [48]. Since the production of fluorinated natural products is extremely rare [49] we analysed the genome of S07_1.15 for the presence of a fluorinase encoding gene. BlastP analysis retrieved no hit for a FlA4 homologue and MAUVE alignment showed that the genomic region of *S. xinghaiensis* harbouring the gene cluster responsible for the biosynthesis of the fluorometabolite in *S. xinghaiensis* shows low synteny between the two strains.

## 3. Discussion

In this work, we report the high-quality de novo sequencing, assembly and genome mining of two new *Streptomyces* isolates from deep-sea samples. Like their terrestrial counterparts, marine Actinobacteria are known to be a valuable source of novel bioactive metabolites mainly due to their rich and mostly unexplored secondary metabolism [50,51]. Identifying and sequencing novel species increases the repository of known biosynthetic gene clusters (BGCs), which constitutes a valuable resource for natural product discovery. 

The number of high-quality *Streptomyces* genomes assemblies available at the NCBI database is less than 15% of the total *Streptomyces* genomes available [52]. Bacterial genome mining for BGCs identification requires high-quality genome assemblies to guarantee sequence continuity. By combining short- and long-read sequencing methodologies, we have obtained high-quality genome assemblies for the two isolates. Indeed, the high percentage (over 99%) of core genes identified confirms the quality and completeness of both assemblies [24,53]. Nevertheless, the genome assembly of isolate S07_1.15 originated two contigs with different average coverages. The higher coverage of the 160 kb contig suggests either a duplicated chromosomal region or the presence of an extrachromosomal replicon. However, the analysis of the genes annotated in the 160 kb contig revealed only three putative plasmid related genes. A MAUVE alignment of S07_1.15 assembly with the closely related S. *xinghaiensis* S187 aligns the 160 kb contig with the 3’ region of the S187 genome, suggesting that this contig might correspond to the 3’ region of strain S07_1.15. However, the increased coverage indicates that this region might be duplicated in the genome and could correspond to the terminal inverted repeats. Nonetheless, no putative BGCs were identified in this genomic region. 

The rapidly increasing availability of whole-genome sequences has claimed for the definition of genomic-based taxonomic metrics that together with genome-wide phylogeny, would support the definition of species in the genomic era [31]. In this context, it is generally accepted that a new species should present a 16S similarity lower than 98.7% and/or ANI (average nucleotide identity) and dDDH (digital DNA–DNA hybridization) values below the thresholds of 95–96% and 70%, respectively [31,54]. The phylogenetic analysis carried out in this work indicated that both isolates belong to the *Streptomyces* genus. In the 16S rRNA, MLSA and WGS phylogenetic trees, isolate S07_1.15 consistently clustered together with *S. xinghaiensis* strains, suggesting that it belongs to the same species. In addition, genome mining of isolate S07_1.15 showed a very similar BGC profile to *S. xinghaiensis* S187 [29]. Despite the resemblance between these two strains, there are a few differences that support the potential for the production of new NPs (Appendix A). Interestingly, unlike S. *xinghaiensis* S187, isolate S017_1.15 does not present a fluorinase-encoding gene. A MAUVE alignment between strains S187 and S017_1.15 shows a lack of synteny in this particular genomic region, which might suggest a recent gene loss or gain event of the fluorinase gene.

Concerning isolate MA3_2.13, this strain did not cluster consistently with other *Streptomyces* strains in any of the phylogenetic analyses performed. Furthermore, the ANI values obtained were below the threshold recommended for prokaryotic species delineation. These results support the claim that isolate MA3_2.13 is a new *Streptomyces* species. The higher proportion of genes assigned to metabolism-related COG categories, the high number of putative novel BGCs identified and the high percentage of genome devoted to secondary metabolism, in comparison to other bacterial species [36,55], all point to a promising strain for obtaining novel chemical structures of pharmaceutical relevance.

Both newly sequenced genomes contain several genes belonging to the MAG lists previously identified for *Salinospora* and *Streptomyces* species [33,34,35], which is consistent with their isolation from deep-sea samples. Several putative genomic islands were also identified, which is common in deep-sea bacteria [56], with the majority of islands containing features related to genomic instability (e.g., transposases). Additionally, putative prophage regions were identified in both genomes, and in the case of MA3_2.13, the identified region is flanked by *attL* and *attR* sites.

Overall, we provide high-quality genome sequences of two deep-sea isolates and determine their biosynthetic potential for the production of novel NPs. In silico dereplication showed that strain MA3_2.13 displays a high potential for production of novel chemical structures and would merit thorough analysis of broth extracts for new NPs or even heterologous expression of the most promising BGCs.

## 4. Materials and Methods

### 4.1. Sampling, Isolation and Microbial Growth

Deep-sea sampling surveys in Madeira archipelago were undertaken in the scope of two oceanographic expeditions. Isolate MA3_2.13 was obtained from sediment collected at 2300 m depth (32.52188 N 16.96831 W) during the SEDMAR 1/2017 mission, with a Box-Corer. Isolate S07_1.15 was retrieved from a sponge (*Demospongiae* sp.) collected at 650 m depth (32.64812 N 17.090578 W) during the OOM-2018 campaign, with the ROV LUSO 6000/EMEPC. Cultivable microorganisms from these deep-sea samples were obtained following a protocol specific for Actinobacteria. Briefly, the deep-sea sample that led to the isolation of strain MA3 2.13 was subjected to a pre-treatment that consisted of incubating 1 g of sediment in a water bath at 57 °C for 15 min, while the deep-sea sponge sample from which strain S_071 1.15 was retrieved was macerated and subjected to a pre-treatment consisting in adding (to 1 g of macerated sponge) 1 mL of seawater and 20 mg/L of nalidixic acid, cycloheximide and nystatin and incubating at room temperature for 30 min. After the incubation period, the samples were ten-fold diluted until 10^−3^ and an aliquot of 100 μl of each dilution was spread over the surface of two selective culture media: M1 agar (composition per liter of seawater: 10 g of soluble starch, 4 g of yeast extract, 2 g of peptone and 17 g of agar) and M4 agar (composition per liter of seawater: 2 g of chitin and 17 g of agar), supplemented with cycloheximide (50 mg L^−1^), nalidixic acid (50 mg L^−1^) and nystatin (50 mg L^−1^). The plates were incubated for up to 6 months at 28 °C. Axenic cultures of each isolate were obtained by repetitive streaking of individual colonies on new agar plates. Each isolate was cryopreserved at −80 °C in 30% (v/v) glycerol. For spore production, Isolates MA3_2.13 and S07_1.15 were grown at 30 ºC on Difco ISP4 solid medium (BD, Franklin Lakes, NJ, USA).

### 4.2. Genomic DNA Isolation and PCR Amplification

Genomic DNA of both isolates was extracted with the E.Z.N.A.^®^ Bacterial DNA Kit (Omega Bio-Tek, Norcross, GA, USA), following the manufacturer’s instructions with a few modification steps: (i) before starting the extraction protocol, samples were incubated at 95 °C for 10 min, followed by incubation on ice for 10 min; (ii) in the step of lysozyme addition, the samples were incubated at 37 °C for 30 min instead of 10 min as described in the protocol; (iii) in the optional step used for Gram-positive bacteria, two Zirconia beads (2.3 mm diameter) were added together with the glass beads and the samples were vortexed for 10 min; (iv) incubation with proteinase K was performed by addition of a concentrated stock (10 mg mL^−1^) instead of the solution provided in the kit and was extended up to 2 h (v) the centrifugation speeds described in the kit protocol were increased in all steps from 10,000 g to 13,000 g. 16S rRNA gene was amplified by PCR using the universal primers 27F (5′-GAGTTTGATCCTGGCTCAG-3′) and 1492R (5′-TACGGYTACCTTGTTACGACTT-3′) [57]. The PCR reaction (total volume 10 µL) contained 5 µL of Taq PCR Master Mix (Qiagen, CA, USA), 0.2 µM of each primer and 3 μL of DNA template. The PCR conditions included an initial denaturation at 95 °C for 15 min, followed by 30 cycles of 30 s at 94 °C, 90 s at 48 °C and 90 s at 72 °C; and a final extension at 72 °C for 10 min. Purification and sequencing of the DNA was performed at GenCore platform (I3S—Instituto de Investigação e Inovação em Saúde, Porto, Portugal). 

Genomic DNA isolation for whole-genome sequencing was carried out using the GeneJET Genomic DNA Purification Kit (Thermo Fisher Scientific, Waltham, MA, USA). Liquid cultures were grown in TSB media at 30 °C, with aeration (220 rpm) for 48 h (isolate MA3_2.13) or 72 h (isolateS07_1.15). Cells from 50 mL cultures were harvested by centrifugation, washed with TE buffer and genomic DNA was extracted according to the manufacturer’s instructions. The quality and quantity of extracted DNA were evaluated by gel electrophoresis and Nanodrop (Thermo Fisher Scientific).

### 4.3. Short-Read (Illumina) and Long-Read (PacBio) Sequencing

Illumina and PacBio library preparation and sequencing were performed at Novogene (Cambridge, UK). PCR-free Illumina sequencing libraries (average insert size of 350 bp) were generated using NEBNext Ultra II DNA Library Prep Kit for Illumina (New England Biolabs, Ipswich, MA, USA), following manufactures’ recommendations. DNA libraries were paired-end sequenced (2 × 150 bp) in a NovaSeq 6000 sequencer (Illumina, Ipswich, CA, USA). Raw data were filtered for high-quality adapter-free reads for genome assembly (cut-off Q score, 5). Genomic DNA from both isolates was also used for the construction of a SMRTbell library and sequenced on a PacBio Sequel system (Pacific Biosciences, Menlo Park, CA, USA). 

### 4.4. De Novo Genome Assembly and Annotation

Short-reads (Illumina) and long-reads (PacBio) were assembled with the hybrid pipeline implemented in Unicycler (v. 0.4.9b) [9] with default software parameters and switches “--mode normal --threads 8”. Manual curation of the assemblies was made based on the mapping of the quality-filtered Illumina paired-end reads to the Unicycler assembly using Bowtie 2 (v. 2.3.2) [58] implemented in Geneious Prime (Biomatters, Auckland, New Zealand). Conflicts showing more than 80% frequency for short reads were corrected according to the Illumina assembly consensus. 

Final assemblies were annotated using RAST (Rapid Annotation using Subsystem Technology) server version 2.0 [22] with the default software parameters (taxonomy NCBI ID: 1883). For submission to the GenBank database, genome annotation was performed using the NCBI Prokaryotic Genome Annotation Pipeline (PGAP) [59]. Prediction of specialized metabolites biosynthetic gene clusters (BGC) was performed with antiSMASH 5.0 using strict detection [23], BAGEL4 [60] and RiPPMiner [61] specifically for RiPPs; and NRPSpredictor2 [62] for NRPS. Functional annotation of predicted gene products of strains MA3_213 and S07_1.15 was carried out using eggNOG-mapper v2 [63,64], using the default parameters (minimum hit e-value 0.001, auto adjust per query the taxonomic scope and transfer annotations from any ortholog). Hits to each COG category were retrieved, with multi-COG hits added to each individual category. Results were normalized as a percentage, using the total number of proteins in each genome. For comparison purposes, the same procedure was carried out for *S. coelicolor* A3(2), *S. avermitilis* MA4680 and *S. griseus* NBRC 13350. Genomic island prediction was carried out using the IslandViewer software [65] and annotation of prophage sequences was performed using the PHASTER web server [66], using the default settings.

### 4.5. Identification of Putative Marine Adaptation Genes

For the identification of the presence of putative MAGs in the two sequenced genomes, the MAG lists of three previous works [33,34,35] were retrieved. In the case of the work by Almeida et. al [35], in order to correctly retrieve the identified genes, the pangenome analysis was recreated using the EDGAR 3.0 platform [67]. A blastp analysis was carried out with the corresponding protein sequences of a total of 207 genes (which included each identified MAG and identified orthologs in the original works) against a local database containing the two newly assembled genomes. BLAST threshold for a positive hit was set as: query coverage higher than 50%, E-value lower than 1e-15 and % identity higher than 35%. In the cases where blastp hits were below the threshold, but the RAST annotation was indicative of the putative MAG searched, the results were considered as a positive hit (identified as annotation only hits)

### 4.6. Phylogenetic Analysis

For molecular-based identification of isolates MA3_2.13 and S07_1.15, the corresponding 16S rRNA sequences were retrieved from the final assemblies. The top 20 hits against each isolate (source: isolates) from the Ribosomal Database Project (RDP release 11) [68], together with the 16S rRNA sequences available at the pubMLST *Streptomyces* database [69] and the top 100 blastn results against the nr and wgs databases, were retrieved and manually curated (for a total of 683 sequences). A maximum likelihood tree was constructed, using the general time reversible model (GTR+G+I) and a bootstrap analysis of 1000 replicates using Mega X [70]. The published sequence for *Kitasatospora setae* KM-6054 (accession number: NC_016109.1) [71] was used as an outgroup. 

To complement 16S rRNA identification, an MLSA analysis was carried out based on the MLST scheme for *Streptomyces* available at pubMLST [69]. Sequences for the *atpD*, *gyrB*, *recA*, *rpoB* and *trpB* genes were retrieved from the final MA3_2.13 and S07_1.15 genome assemblies and the top blastn hits (nr/nt and WGS databases) for each gene were also compiled and curated. Sequences were concatenated and a maximum likelihood tree, using the general time reversible model (GTR+G+I) and a bootstrap analysis of 2000 replicates, was constructed using a total of 300 sequences in Mega X [70]. The published sequences for the five genes of *Kitasatospora setae* KM-6054 were used as an outgroup.

The genome sequence of the two isolates together with the *Streptomyces* complete NCBI RefSeq assemblies (total of 280 assemblies on March 2021) were used for the construction of a whole-genome-based phylogenetic tree using the GToTree (v. 1.5.47) workflow [30] with IQ-TREE (v. 2.0.3) program for tree generation [72]. The GToTree workflow was implemented with default settings and using the single-copy gene set of 138 target genes specific for Actinobacteria. *Kitasatospora setae* KM-6054 was used as an outgroup. Whole-genome average nucleotide identity was calculated with PYANI (v. 0.2.10) module [73] using the ANIb algorithm. Additionally, to gain further insights on isolate identification, both assemblies were analysed using the KmerFinder (v. 3.2) software [74] and the Microbial Genomes Atlas Online (MiGA Online) [27].

## Figures and Tables

**Figure 1 marinedrugs-19-00621-f001:**
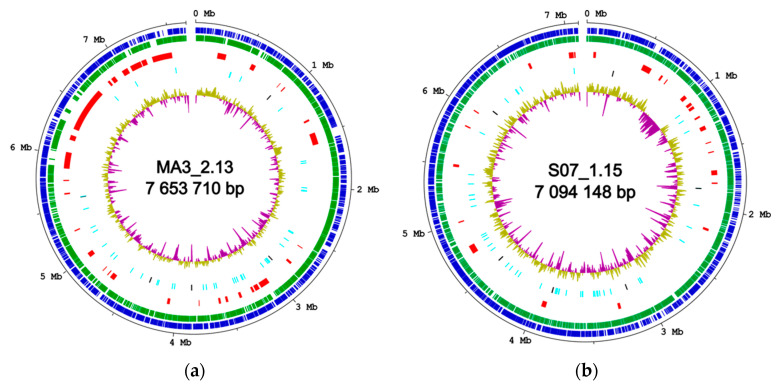
Schematic representation of the chromosomes of isolates MA3_2.13 (**a**) and S07_1.15 (**b**) generated by DNAPlotter v 18.1.0. The chromosomes are represented as open circles and for S07_1.15, only the large contig is shown. From outside to inside, the concentric circles represent: genome coordinates, coding sequences (CDS) in the forward strain (in blue) and in the reverse strain (in green), regions of putative BGCs (in red), tRNA and rRNA genes (in cyan and in black, respectively); GC percentage plot with default settings (above average in olive and below average in purple).

**Figure 2 marinedrugs-19-00621-f002:**
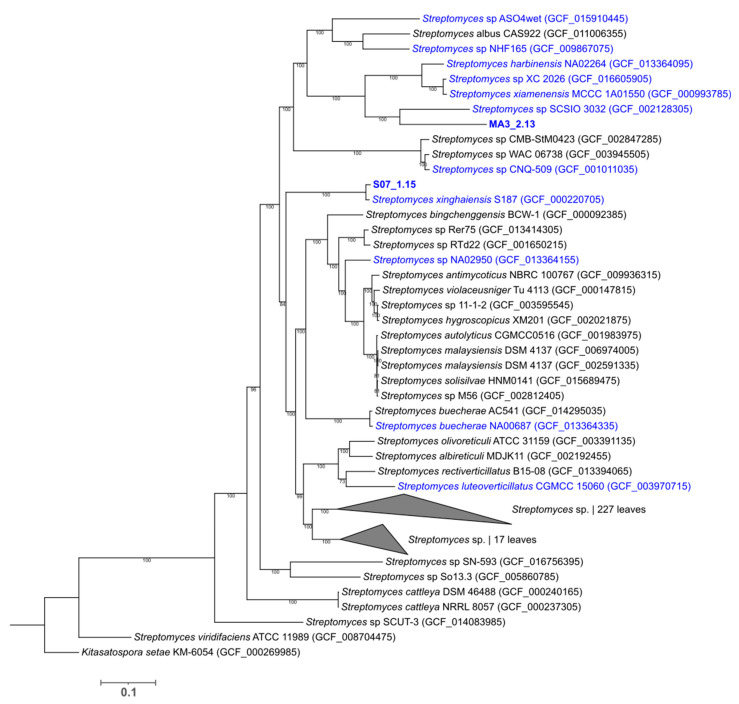
WGS phylogenetic tree of 280 NCBI RefSeq *Streptomyces* strains and isolates MA3_2.13 and S07_1.15 (highlighted in bold), generated using the GToTree workflow and visualized with the web-based tool Interactive Tree of Life (https://itol.embl.de/ (accessed on 10 October 2021)). Portions of the tree collapsed are labelled and numbers represent the number of leaves/genomes in the collapsed subtrees. Strains name in blue indicate strains isolated from marine samples.

**Figure 3 marinedrugs-19-00621-f003:**
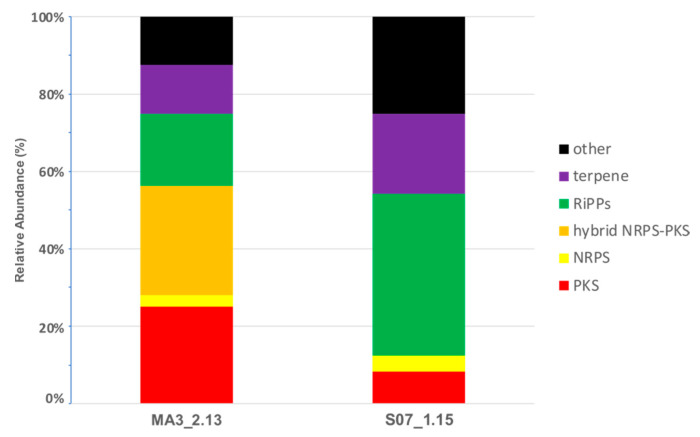
Occurrence of BGCs types in both strains as predicted by antiSMASH. Data were retrieved from Appendix A.

**Table 1 marinedrugs-19-00621-t001:** General features of the genome sequence of isolates.

Isolate	Genome Size (bp)	Fold Coverage (x)	G+C Content (%)	No. of CDS ^1^	No. of rRNA Operons	No. of tRNA Genes	No. of BGCs ^2^	GenBank Accession Number
MA3_2.13	7,653,710	139	72.1	6412	5	55	32	CP082362
S07_1.15	7,094,148	159	73.2	6492	6	62	24	JAJBZK000000000
	160,397							

^1^ CDS—coding DNA sequences. As determined through RAST automatic annotation [22]. ^2^ BGCs—biosynthetic gene clusters determined through antiSMASH [23].

## Data Availability

The fully complete genome sequences were deposited in the NCBI GenBank (BioProject ID PRJNA754006) under accession numbers CP082362 for isolate MA3_2.13 and JAJBZK000000000 for S07_1.15.

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
