# Peer review of "Complete Genome Sequence of Two Deep-Sea Streptomyces Isolates from Madeira Archipelago and Evaluation of Their Biosynthetic Potential"

_marinedrugs, 2021, doi:10.3390/md19110621_

Round 1

Reviewer 1 Report

The authors have sequenced the genome of two marine actinomycetes, collected from the Madeira archipelago, and analyzed the natural product production potential of these organisms using computational methods. Detailed phylogenetic analysis is also presented. Multiple putative PKS, NRPS, PKS-NRPS hybrid, and RiPP biosynthetic gene clusters were identified. Results are presented clearly and the findings should be of interest to some researchers. Clearly, a lot more work needs to be done to find out if either, or both, of these organisms produce natural products that can be used as human medicine. But the genome information contained in this manuscript is of sufficient value to warrant publication (in present form).

Author Response

The authors thank the reviewer for the comments. We agree that further “wet lab” work will be required to characterize the putative novel compounds produced by these organisms. It is our intention to pursue this line of work in the future. However, we do share the same opinion as the reviewer, that the information reported in this manuscript is a good foundation to guide further discovery studies, both in the “in silico” and “wet lab” stages.

Reviewer 2 Report

This manuscript titled "Complete genome sequence of two deep-sea Streptomyces isolates from Madeira archipelago and evaluation of their biosynthetic potential is well described the detail whole genome information of two isolates of Streptomyces, which have the potential ability to produce the natural products. Authors used hybrid DNA sequencing technologies (Pacbio and Illumina) sequenced the Streptomyces and got the good quality genome of these bacteria including one novel species. However, this article only shows the genome information and prediction of secondary metabolite biosynthesis gene cluster. If the real metabolites information of these bacteria and relationship of metabolites and biosynthesis gene clusters could be obtained and discussion, it will supply more contributions for discovery the marine drugs.  

Author Response

The authors acknowledge the limitations of the paper concerning obtaining and chemically characterizing the putative novel compounds, particularly those produced by isolate MA3. However, the main scope of the paper, as acknowledged by the reviewer, was to provide a robust in silico pipeline, not only for obtaining high quality bacterial genomes, but also to systematize genome mining strategies for novel compounds. It is therefore our hope, that the main contribution of this article is to enhance the in silico discovery phase of new compounds, particularly those produced by deep-sea dwelling bacteria.

Reviewer 3 Report

The manuscript by Albuquerque et al “Complete genome sequence of two deep-sea Streptomyces isolates from Madeira archipelago and evaluation of their biosynthetic potential” deals with the isolation and biosynthetic potential of two deep-sea Streptomyces isolates. The two strains were assigned to the genus Streptomyces, a well-known taxon for its bioactive potential. Although the authors have reported the strains from an interesting location, the data obtained by the authors is not sufficient nor interpreted and presented properly which makes it unsuitable for publication in Marine drugs.

From this reviewer’s perspective, the characterization done by the authors lacks both the experimental and the bioinformatics analysis of the strains. Streptomyces are known not only for their biosynthetic potential but also for other enzymatic activities. For example, basic experiments for their potential to evaluate biodegradation of macromolecules or tests for antimicrobial activities could have enhanced the potential of this work. There is no single experiment that would suggest that these strains may have some biological activity.

Line No: 28-29: The authors propose that strain MA3_2.13 is a new species, but do not provide and show any results for experimental characterization for this strain.

The sections of the manuscript describing the content of BGCs and their predictions are only describing the output of antismash, which is often simplistic, confusing, or indicates obvious observations. For example: in lines 245-257 the authors mention the similarity between the atratumycin BGC cluster from S. atratus SCIO ZH16 and strain MA3_2.13. However they do not show if there are any differences in the modular structure of the 3 NRPS proteins from these two strains. At least the authors could have shown these similarities or differences as a figure. Moreover, what are the sequence similarities between these NRPS sequences from the two mentioned strains? Also, the MIBiG accession number for SCIO ZH16 strain is missing from the manuscript. For this reviewer, the analysis provided in the current manuscript is just a plain description of what anyone interested could see in an output of the AntiSMASH pipeline.

Is any of the natural products found in the two genomes also found in other deep-sea bacteria?

On lines 298-304, the authors mention that isolate S07_1.15 is closely related to S. xinghaiensis S187 but lacks BGC for fluoroacetate. It is not clear if S. xinghaiensis S187 was also reported from the deep-sea. If it’s not, then the isolation source of S. xinghaiensis S187 should be discussed along with the habitats of other fluoroacetate-producing organisms (if any).

Again, section 2.4 on the marine adaptation genes is a simple description based on the output of blastp analysis. No attempt was made to actually experimentally validate the presence of any of these putative genes.

Since the authors already have a phylogenomic-based tree, the phylogenetic analyses based on 16S rRNA and MLST does not provide any additional value, therefore, these should be removed from the manuscript. Moreover, strain names should be italicized in phylogenetic trees.

Additional Comments:

Introduction lacks a report on previously described NPs from deep-sea Streptomyces.

Line No. 458: adaptacion → adaptation

Line Nos. 499-500: The genomes could not be found in the NCBI database.

Supplementary Table S4: Remove the “%” sign from each cell.

In supplementary data, strain S07_1.15 is denoted as S071_1.15 in some places. This should be checked thoroughly (in the main manuscript as well) and fixed.

The authors should look at some of the recent papers on Streptomyces genomes (https://doi.org/10.3390/microorganisms9091802 and https://doi.org/10.3390/md19080440) published in the marine drugs as a reference.

Round 2

Reviewer 2 Report

no extra comments

Author Response

No reply needed.